# Construction of an Elastin-like Polypeptide Gene in a High Copy Number Plasmid Using a Modified Method of Recursive Directional Ligation

Derek W. Nelson [1,2] , Alexander Connor [1,3], Yu Shen [4] and Ryan J. Gilbert [1,2,5,]*

1 Center for Biotechnology and Interdisciplinary Studies, Rensselaer Polytechnic Institute, 1623 15th St., Troy, NY 12180, USA; nelsod4@rpi.edu (D.W.N.); aconnor95@gmail.com (A.C.)
2 Department of Biomedical Engineering, Rensselaer Polytechnic Institute, 110 8th St., Troy, NY 12180, USA
3 Department of Chemical and Biological Engineering, Rensselaer Polytechnic Institute, Troy, NY 12180, USA
4 Department of Chemical and Biomolecular Engineering, Johns Hopkins University, Baltimore, MD 21218, USA; yushen986@outlook.com
5 Albany Stratton Veteran Affairs Medical Center, 113 Holland Ave., Albany, NY 12208, USA
* Correspondence: gilber2@rpi.edu

**Abstract:** Elastin-like polypeptides (ELPs) are popular biomaterials due to their reversible, temperature-dependent phase separation and their tunability, which is achievable by evolving procedures in recombinant technology. In particular, recursive direction ligation by plasmid reconstruction (PRe-RDL) is the predominant cloning technique used to generate ELPs of varying lengths. Pre-RDL provides precise control over the number of $(VPGXG)_n$ repeat units in an ELP due to the selection of type IIS restriction enzyme (REs) sites in the reconstructed pET expression plasmid, which is a low-to-medium copy number plasmid. While Pre-RDL can be used to seamlessly repeat essentially any gene sequence and overcome limitations of previous cloning practices, we modified the Pre-RDL technique, where a high copy number plasmid (pBluescript II SK(+)—using a new library of type IIS REs) was used instead of a pET plasmid. The modified technique successfully produced a diblock ELP gene of 240 pentapeptide repeats from 30 pentapeptide "monomers" composed of alanine, tyrosine, and leucine X residues. This study found that the large, GC-rich ELP gene compromised plasmid yields in pBluescript II SK(+) and favored higher plasmid yields in the pET19b expression plasmid. Additionally, the BL21 *E. coli* strain expression consistently provided a higher transformation efficiency and higher plasmid yield than the high cloning efficiency strain TOP10 *E. coli*. We hypothesize that the plasmid/high GC gene ratio may play a significant role in these observations, and not the total plasmid size or the total plasmid GC content. While expression of the final gene resulted in a diblock ELP with a phase separation temperature of 34.5 °C, future work will need to investigate RDL techniques in additional plasmids to understand the primary driving factors for improving yields of plasmids with large ELP-encoding genes.

**Keywords:** recursive directional ligation; elastin-like polypeptides; recombinant DNA engineering; *E. coli* plasmid yield

## 1. Introduction

Elastin-like polypeptides (ELPs) are biomimetic molecules derived from a structural motif found in tropoelastin. ELPs are popular biomaterials due to their independently tunable properties, including reversible phase separation and gelation, enzymatic degradation, cell adhesion, and therapeutic conjugation [1,2]. Thus, ELPs are used for many biomedical applications, including targeted drug delivery and tissue engineering [3–7], where their tunable properties are leveraged for the development of injectable hydrogels [8–10], nanoparticle formulations [11–14], electrospun fibers [15–17], and bio-inks for 3D bioprinting [18,19]. The development of these various biomaterials is often controlled by variations

of the primary ELP pentapeptide motif $(VPGXG)_n$, where X indicates a tunable residue that can be any amino acid, except for proline, and n indicates the number of pentapeptide repeats. Altering these parameters provides sites for chemical modification or hydrogel crosslinking, facilitates drug interactions, and elicits temperature-triggered gelation or self-assembly into a hierarchical structure [4,20,21]. Additionally, ELPs are often appended with various peptide motifs, such as adhesion or enzyme-degradable sequences [1,22,23], or therapeutic proteins and peptides [24,25]. Therefore, since they can be tailored to provide scaffolding for cell migration and growth or elicit local and targeted drug delivery in a variety of biomedical applications, ELPs are a promising biomaterial substrate.

The precise control of many ELP properties is possible via the use of recombinant technology, where basic molecular biology techniques are employed to modify an ELP-encoding gene to then induce the protein expression of that gene using an *E. coli* host. Due to ELP reversible phase separation, purification after expression is performed using hot and cold centrifugal steps, known as inverse thermal cycling (ITC), to remove soluble and insoluble bacterial contaminants [26]. The ITC method enables the usage of ELPs as purification tags for conjugated therapeutic proteins to offset cost-intensive chromatography in a process known as ELPylation [27,28]. However, even these ELP tags must be tailored to elicit phase separation at lower temperatures so that the degradation of the appended therapeutic does not occur. While there are many basic molecular biology techniques for the integration of adhesion motifs, degradation sequences, and therapeutic peptides or proteins with ELP genes, precise control of ELP pentapeptide length relies on niche techniques, specific to repetitive polypeptides. Initially, Meyer and Chilkoti demonstrated that ELP pentapeptide length could be controlled using a method known as recursive directional ligation (RDL) [29,30]. For ELP biopolymers, RDL was developed to contain compatible type IIP restriction enzyme (RE) recognition sites (PflMI and BglI) encoded onto the ends of the ELP gene, such that the parallel digestion reaction of a pUC19 vector with either one or both enzymes would result in a monomer ELP gene with a linearized pUC19 plasmid or a monomer ELP gene by itself, respectively. These digest products are then ligated together due to compatible sticky ends from the selected REs, resulting in a new pUC19 plasmid containing a dimer of the ELP gene [29]. While RDL provides a means for controlling the number of ELP pentapeptide repeats, the cloning efficiency is compromised by the self-ligation of the monomer ELP gene to itself. The resulting re-circularization of the ELP monomer gene in the pUC19 plasmid results in colonies with the original plasmid construct, with only the monomer ELP gene and tandem repeat ligations of smaller gene inserts (<500 bp). Meyer and Chilkoti report that 30–80% of clones are positive for the intended insert, and that 10–20% of colonies have a double insert when the insert is <500 bp. Additionally, because the RE recognition sites for RDL overlap with the ELP coding region, this system lacks modularity. Indeed, the selected library of REs is specific to only ELP biopolymers, and new libraries of type IIP REs need to be selected to perform RDL with other biopolymers, such as NheI and SpeI for the RDL of recombinant dragline silk biopolymers [30,31]. To address these limitations, McDaniel et al. devised a modified technique termed recursive directional ligation by plasmid reconstruction (PRe-RDL) that utilizes a library of type IIS REs (BseRI and AcuI) whose recognition sequences are integrated in the plasmid before and after the coding region of the ELP gene, but cleave the DNA a precise number of base pairs away from the recognition sequences and in the ELP coding region [32]. This allows for the cleavage of any DNA sequence, as type IIS REs are only specific regarding the cut distance from their recognition site and not the cut sequence. Thus, parallel digest reactions may be carried out so that each reaction has one of these type IIS enzymes and a unique type IIP enzyme (BglI in this study) whose recognition sequence is already encoded in the plasmid, allowing each reaction to yield either a fragment of the plasmid with a 5′ overhang ELP monomer sequence, or the other fragment of the plasmid with a 3′ overhang ELP monomer sequence. When ligating these digest products together, the plasmid fragments are then re-joined, via the sticky ends resulting from BglI, to yield the original full-length plasmid, and the ELP monomers are ligated, through the compatible sticky ends from the type IIS REs to re-circularize the plasmid and hold a dimer of the ELP gene [32]. This method eliminates the possibility of the premature re-circularization of a plasmid with only a

monomer ELP gene, as well as the circularization of a monomer ELP gene to itself. Additionally, because the recognition sites are not encoded in the ELP gene, PRe-RDL is translatable to other biopolymers. Lastly, PRe-RDL was developed in a pET expression plasmid in such a way that every multimer of an ELP gene is ready to be expressed in *E. coli* as is, without requiring an additional digest reaction to move the ELP multimer gene from a non-expression plasmid, such as pUC19 used for RDL, to a pET expression plasmid. This is significant for ELP biopolymers as libraries of ELPs with varied pentamer repeats are often desired to screen for the optimal transition temperature, which is a function of pentapeptide length [2,33].

In this article, we aimed to produce an ELP gene that was composed of a previously published ELP polymer with already-defined properties, but modified with an appended hydrophilic ELP block. Thus, we developed a novel plasmid engineering scheme to use methods similar to PRe-RDL in a high copy number plasmid, pBluescript II SK(+), which lends the possible benefit of providing higher plasmid yields from bacterial cultures. Additionally, the scheme shown in this article utilizes REs from the FastDigest enzyme product line, which facilitates a faster mutagenesis workflow when compared to previously published PRe-RDL methods [32]. Because only a final ELP gene is desired, performing PRe-RDL in a pET expression plasmid was not vital as only the final product would need to be expressed, instead of each incremental increase in ELP gene length. In contrast, this method has limitations in its application to other biopolymers and will always require an additional step to move the final ELP multimer gene from pBluescript SK(+) to a pET plasmid. Following the method established in this study, we also investigate the effect of the GC-rich ELP gene on plasmid yields, resulting from gene size, plasmid selection, and bacterial strain. This investigation provides some insights into factors that affect ELP plasmid yields from bacterial cultures. Lastly, the final ELP gene produced through the novel RDL methodology is used to recombinantly express a diblock ELP with a physiologically relevant transition temperature after purification using ITC.

## 2. Results

### 2.1. ELP Gene Design

To produce an ELP with 240 pentamers, we structured our ELP "monomers" into 30 pentamers or $[(VPGAG)_2\text{-}(VPGXG)_1\text{-}(VPGAG)_2]_6$, and the nomenclature is simplified into ELP-$Y_1$ and ELP-$L_1$, where X = Y or L amino acids. Thus, RDL was performed $2\times$, where the first round used these monomers to produce the dimers ELP-$Y_2$ and ELP-$L_2$, and the second round used these dimers, resulting in the tetramers ELP-$Y_4$ and ELP-$L_4$. A third round of RDL was carried out to combine the tetramers into the final octamer ELP-$L_4Y_4$. To make PRe-RDL possible in Bluescript SK(+), we followed the six design criteria identified by McDaniel et al. to design our ELP monomer genes and select our REs. This resulted in the selection of type IIS REs BpiI and BmsI and type IIP RE AdeI. We also wanted NcoI and BamHI REs to be used for the cloning of the final ELP gene into a pET expression plasmid. PRe-RDL using these REs in the pBluescript SK(+) plasmid was possible by using the following gene design criteria:

1. Include a NcoI RE site (cc atg g) at the beginning of the ELP gene and a BpiI RE site (g tct tc) at the end of the ELP gene.
2. Use a tag stop codon to overlap with the BpiI RE site by one base pair.
3. Include a G amino acid (encoded by ggc) before the first VPGXG and overlap with the NcoI site by 1 base pair.
4. XG in the last VPGXG repeat must be encoded by yyg ggc, where y can be any base.
5. Include a BmsI RE site (gc atc) one base pair before the NcoI RE site and a BamHI restriction enzyme site (g gat cc) after the BpiI site.

These criteria were used to design ELP-$Y_1$ and ELP-$L_1$ genes using SnapGene software version 7.1.2. An XbaI RE site was also added before the BmsI RE site, but is not specified above as it does not specifically play a role in the RDL method. These genes were then synthesized and cloned into pBluescript SK(+) plasmids via XbaI and BamHI REs through Genscript's custom gene synthesis service. Figure 1 depicts the combination of all these

design criteria and demonstrates the methodology used in this study. Because ELP genes are inserted at XbaI and BamHI, there is technically no modification of the plasmid before gene insertion. Additionally, the BpiI RE site overlaps with the stop codon of the ELP gene. Thus, the nomenclature of PRe is inaccurate in this instance since the RE sights must be incorporated into each new gene rather than the plasmid, though the methodology is the same as previously published PRe-RDL [32]. Therefore, this method will be referred to as RDL following gene termini modifications or GTMs-RDL.

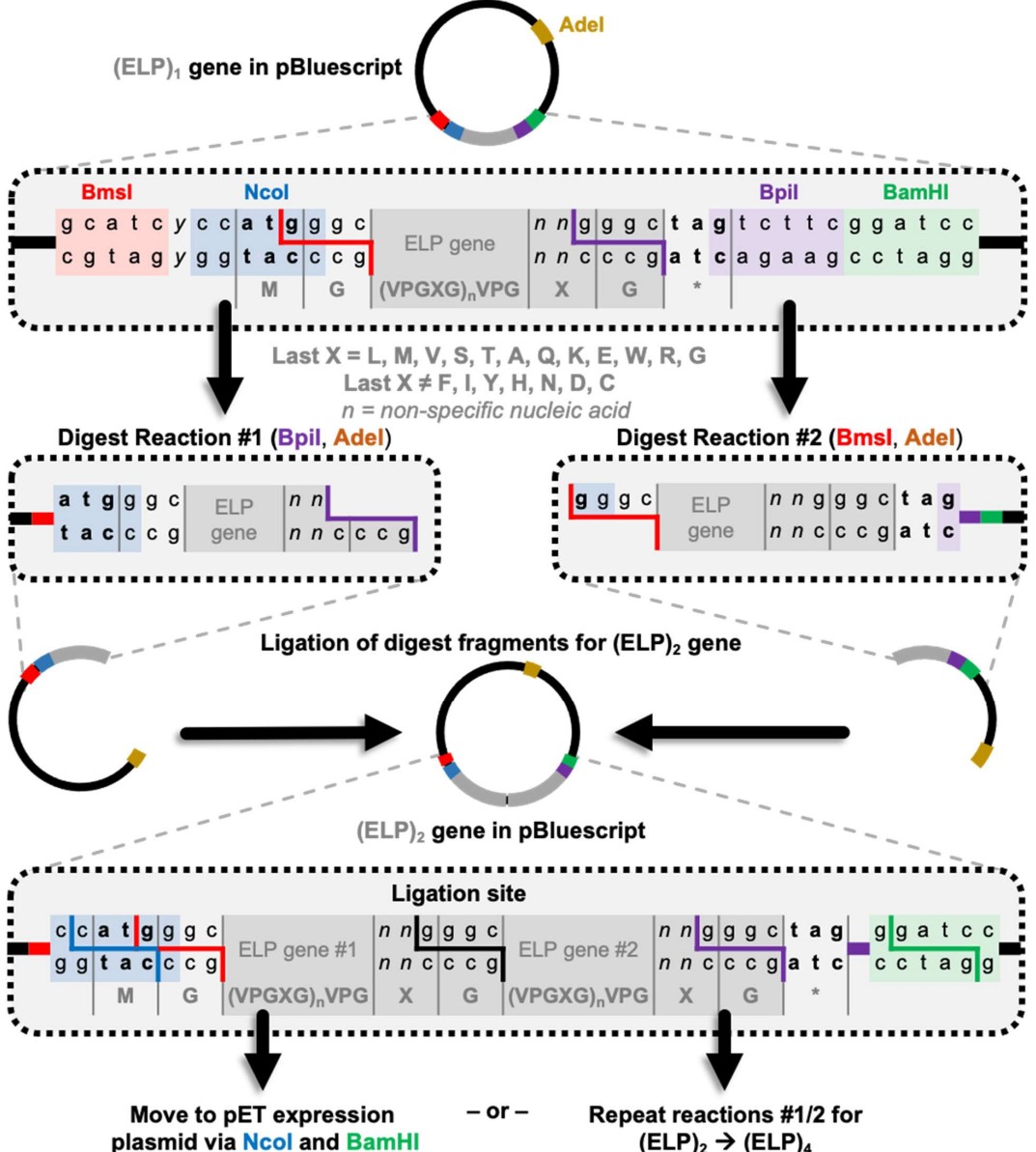

**Figure 1.** Schematic of the GTMs-RDL methodology used in this study. A pBluescript II SK(+) plasmid containing an ELP gene undergoes two parallel digest reactions to create two gene/plasmid fragments of interest. Due to the design of the original ELP gene insert, ligating these two fragments together results in doubling the ELP gene size without a scar sequence in the original pBluescript SK(+) plasmid. This can be repeated as needed or moved to an expression plasmid once the final gene size is achieved. Not shown is the presence of the XbaI restriction enzyme site before the BmsI site to facilitate gene insertion into the pBluescript II SK(+) plasmid. The ELP gene is shown in gray. The enzyme binding sites are highlighted and enzyme cut sites are outlined in brown for AdeI, red for BmsI, blue for NcoI, purple for BpiI, and green for BamHI. * indicates the stop codon sequence (tag).

*2.2. GTMs-RDL Can Be Used to Produce the ELP-$L_4Y_4$ Gene from Monomer Genes in pBluescript II SK(+)*

Using the five design criteria outlined in Section 2.1, ELP-$Y_1$ and ELP-$L_1$ genes were synthesized and inserted into pBluescript II SK(+) plasmids using Genscript's molecular biology services. Following the five general steps outlined in Section 4.1, GTMs-RDL was carried out following Figure 1 to yield ELP-$Y_2$ and ELP-$L_2$ dimer genes, which were then used to produce ELP-$Y_4$ and ELP-$L_4$ tetramer genes in a second round of GTMs-RDL. One last round of GTMs-RDL was performed with both tetramer genes to yield the di-block octamer gene, ELP-$L_4Y_4$. Figure 2A indicates the bands of interest from both GTMs-RDL reactions of monomer, dimer, and tetramer ELP-L genes in pBluescript II SK(+). In this representative agarose gel image, reaction #2 of the monomer plasmid results in overlapping bands without the use of PagI, as seen in lane 4. Therefore, PagI was used (not shown) in reaction #2 of both ELP-$L_1$ and ELP-$Y_1$ genes. The gene libraries after GTMs-RDL are shown in Figure 2B, where pBluescript II SK(+) plasmids were digested with NcoI + BamHI and run through an agarose gel to show the progression for the synthesis of ELP-$L_4Y_4$. This gel demonstrates the success of the GTMs-RDL method in a high copy number plasmid. Once again, the goal of this method was to produce a final ELP peptide and not a library of ELPs with varied sizes. Thus, GTMs-RDL was appropriate for this goal as only one final ELP gene was transferred to a pET expression plasmid, as shown in lane 8 of Figure 2B.

While Figure 2 demonstrates the success of the GTMs-RDL method, a couple of challenges and drawbacks should be mentioned. First, the final octamer gene had a significantly low transformation efficiency in pBluescript II SK(+) after ligation when transforming into TOP10 *E. coli* compared to the monomer, dimer, and tetramer genes. The ligation reaction for the octamer gene was modified to proceed overnight at 4 °C after ~1 h of incubation at room temperature, and *E. coli* was heat shocked with the octamer-containing plasmid for 45 s instead of 30 s, the time used for transforming all other ELP genes into *E. coli*. Together, these modifications to the protocol resulted in colonies containing the final ELP gene. Our second challenge is shown in Figure 3A, where the plasmid yield significantly decreased after every iteration of GTMs-RDL. This observation was not dependent on the variation of the ELP genes encoding Y or L guest residues. By the time we had our octamer gene, a single miniprep of 4 mL of bacterial culture (16 h culture) barely provided enough plasmid for a single RE digest; we desired a minimum of 500 ng of plasmid for every digest reaction. This observation initially implies that either the plasmid size or length of the GC-rich ELP gene was compromising plasmid yields. Lastly, while the goal of this modified RDL method was to acquire a final ELP gene in a high copy number plasmid, pBluescript II SK(+) resulted in low plasmid yields when compared to the final gene in pET19b, a medium copy number plasmid. Therefore, we carried out an additional study to determine the effects of plasmid type and bacterial strain on plasmid yield with the ELP-$L_4Y_4$ gene. Thus, pET19b was transformed into TOP10 *E. coli*, and pBluescript II SK(+) was transformed into BL21 *E. coli* following step 4 from Section 2.2. It should be emphasized that these transformations (Figure S2) used purified plasmids, and not recently ligated plasmids. Step 5 was then performed with these transformed strains as well as the original transformed strains (pBluescript II SK(+) into TOP10 and pET19b into BL21), resulting in the plasmid yields shown in Figure 3B. When looking at just pBluescript II SK(+), the BL21 expression strain provided a plasmid yield that was almost three times higher (2091 ± 106 ng) than TOP10 (752.5 ± 371.4 ng), though this was not statistically significant ($p < 0.078$), indicating that GTMs-RDL should be carried out in the expression strain instead of the maintenance strain. It should also be noted that TOP10 transformation efficiency, based on the number of colonies following transformation, was lower than the BL21 transformation regardless of plasmid type (example shown on ampicillin-resistant plates in Figure S2), demonstrating another benefit to using the expression strain instead of the maintenance strain with this size of ELP. This observation, combined with the low transformation efficiency of the ligated ELP-$Y_4L_4$ plasmid into TOP10 after GTMs-RDL mentioned above, implies that ligation

efficiency may not be the obstruction to transformation efficiency, but rather the bacterial strain may play a larger role in the transformation of this large, GC-rich gene. The PRe-RDL method also relied on TOP10 *E. coli* for transformations, but no comparison was made to the BL21 expression strain [32]. Additionally, Figure 3B shows that inserting the ELP-$L_4Y_4$ gene into the pET19b expression plasmid resulted in higher plasmid yields when compared to pBluescript II SK(+) regardless of bacterial strain (9.6× higher in BL21 and 5× higher in TOP10), though BL21 (11,031 ± 1198 ng) resulted in higher yields when compared to TOP10 (7234 ± 478.8 ng). This finding indicates that plasmid size was not a factor for plasmid yield since the pET19b plasmid consisted of 9254 base pairs and the pBluescript plasmid consisted of 6612 base pairs (including the octamer ELP gene). This is surprising not only because both plasmids have the same origin of replication (yellow arrow labeled "ori" in Figure 3C,D), but also because the pET19b plasmid expresses the repressor of primer (purple arrow labeled "rop" in Figure 3D) proteins, which should reduce the copy number of pET plasmids relative to plasmids lacking this gene [34,35]. Additionally, there was no difference in cell density after 16 h, as indicated by OD600 measurements in Figure S3, which implies that the plasmids did not significantly affect bacterial growth. pBluescript II SK(+) went from 50% total GC content to 65% after inserting the ELP-$L_4Y_4$ gene, and pET19b moved from 54% total GC content to 63% after gene insertion. This shows that the total GC content was not drastically different between plasmids. However, Figure 3C shows how large this ELP gene is compared to both the pBluescript II SK(+) plasmid (55% of the whole plasmid) and its high GC content. Figure 3D shows the same gene but in pET19b, where it comprises a smaller percentage (39%) of the plasmid.

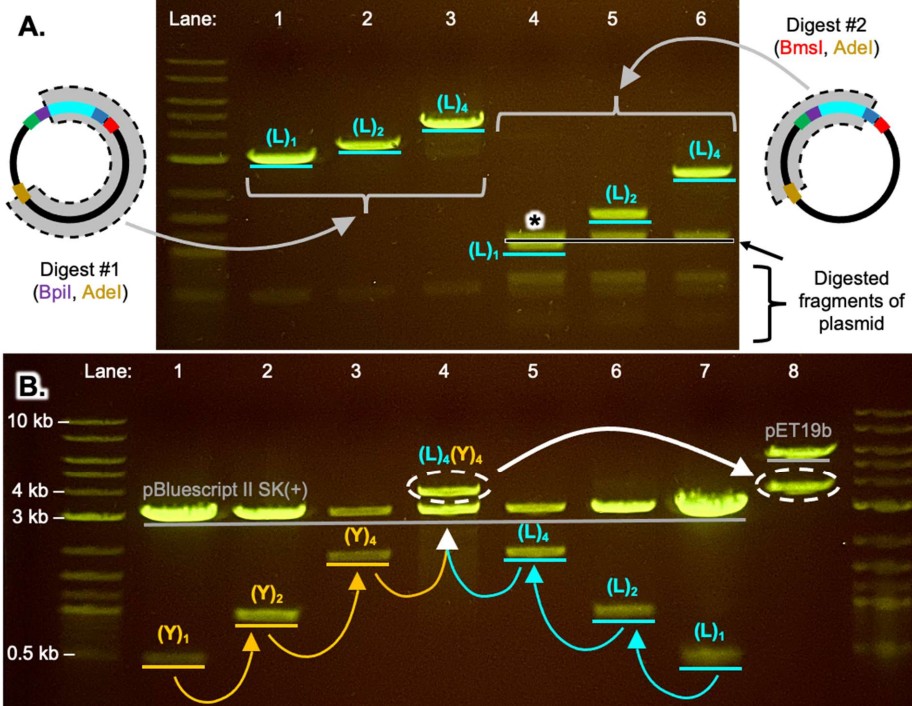

**Figure 2.** Representative images of agarose gels demonstrating the bands of interest during GTMs-RDL (**A**) and the confirmation of the ligation of those bands of interest in pBluescript II SK(+) for the final synthesis of the ELP-$L_4Y_4$ gene (**B**). For example, bands of interest from lanes 1 and 4 in (**A**) were ligated together to produce the ELP-$L_2$ gene shown in lane 6 of (**B**). NcoI and BamHI restriction enzymes were used in (**B**) to remove each ELP gene from its plasmid. The white arrow in (**B**) indicates the successful ligation of the ELP-$L_4Y_4$ gene in lane 4 from pBluescript II SK(+) to pET19b in lane 8. Orange indicates ELP genes encoding tyrosine residues and cyan indicates ELP genes encoding leucine residues. * indicates a scenario where a PagI restriction enzyme is needed to remove overlapping bands with the band of interest.

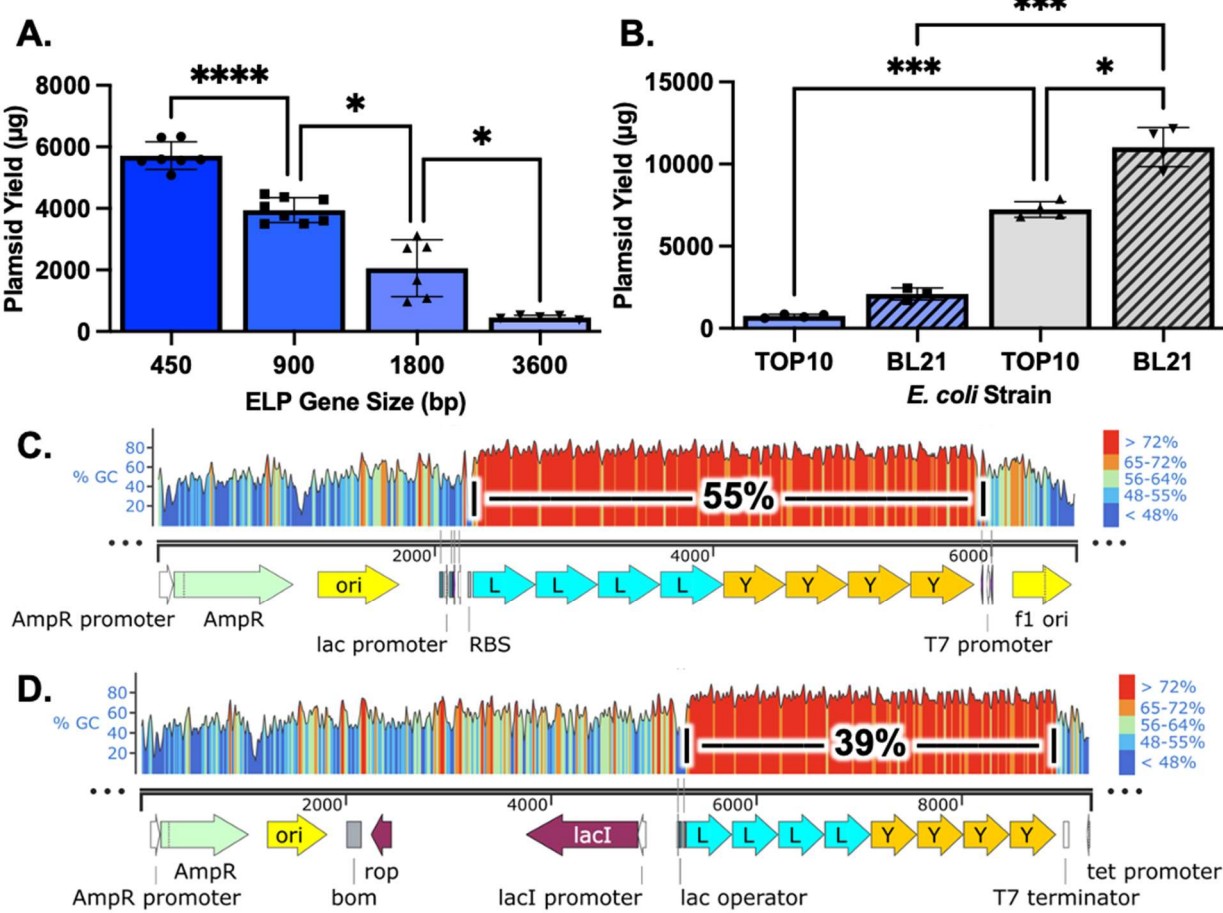

**Figure 3.** ELP plasmid yield is dependent on gene size, plasmid selection, and bacterial strain. Increasing ELP gene size correlates with decreased plasmid yields (**A**). Gene sizes correlate with monomer, dimer, tetramer, and octamer ELP genes in this study, left to right on the x-axis (**A**). Additionally, octamer pET19b plasmids (gray) resulted in higher yields compared to octamer pBluescript II SK(+) plasmids (blue), and BL21 *E. coli* also improved plasmid yields compared to TOP10 *E. coli* strains (**B**). Lastly, the final ELP octamer gene (cyan and orange arrows labeled L, Y) is shown in pBluescript II SK(+) (**C**) and pET19b (**D**) to represent GC content and the ELP gene size relative to each plasmid as a whole. Brown–Forsythe and Welch ANOVA tests were performed for both (**A**,**B**). Statistical significance is indicated by * ($p < 0.05$), *** ($p < 0.001$), or **** ($p < 0.0001$).

### 2.3. Expression, Purification, and Characterization of ELP-$L_4Y_4$

As seen in Figure 2B, the final octamer gene was cut out of pBluescript II SK(+) via NcoI + BamHI REs and inserted into a pET19b expression plasmid. This plasmid was transformed into BL21 *E. coli* and expression was induced using IPTG in 1 L LB and TB cultures. ELPs were purified using three rounds of ITC, were dialyzed against 18MΩ $H_2O$, and the purity was assessed via a 2100 Bioanalyzer System, as seen in Figure 4A,B. The bioanalyzer uses a high-sensitivity electrophoretic chip that can quantify protein size and purity down to the picogram level using a fluorescent dye; however, like a PAGE gel [26], peptide migration is dependent on the amino acid sequence, and the migrating ELP band often appears larger than its theoretical size. For ELP-$L_4Y_4$, the theoretical MW is 94.97 kDa, whereas the bioanalyzer shows a MW of 138.83 ± 0.71 kDa in Figure 4B. Additionally, Figure 4A shows that three rounds of ITC resulted in an average purity of 95 ± 8% for four separate batches of ELP-$L_4Y_4$ based on an area under the curve (AUC) analysis. However, it should be noted that expressions in TB (lanes 3 and 4) resulted in lower purities when compared to LB media (lanes 1 and 2), though more batches would need to be assessed in order to determine this significance. While LB cultures resulted in no

detectable impurities (100% purity via AUC), LB batches averaged 6.25 mg/L of expression media, whereas TB cultures averaged 9.85 mg/L after dialysis and freeze-drying. Figure 4A also demonstrates these differences in yield based on the fluorescence intensity of the ELP peak, though the bioanalyzer was not calibrated in a way that allowed FU to accurately correlate to concentration. Supplemental Figure S4 shows the bioanalyzer reports for these four batches, where each impurity can be more easily seen and quantified for total protein content by AUC%. Together, these findings indicate that while TB media may provide a higher yield, additional rounds of ITC should be considered to achieve higher purity. It should also be noted that because ELP-$L_4Y_4$ lacks amino acids with primary amino- and thiol- groups, the bioanalyzer is likely underestimating peptide purity as contaminants may have a higher fluorescent signal than the ELP due to the presence of more fluorescently labeled amino acids.

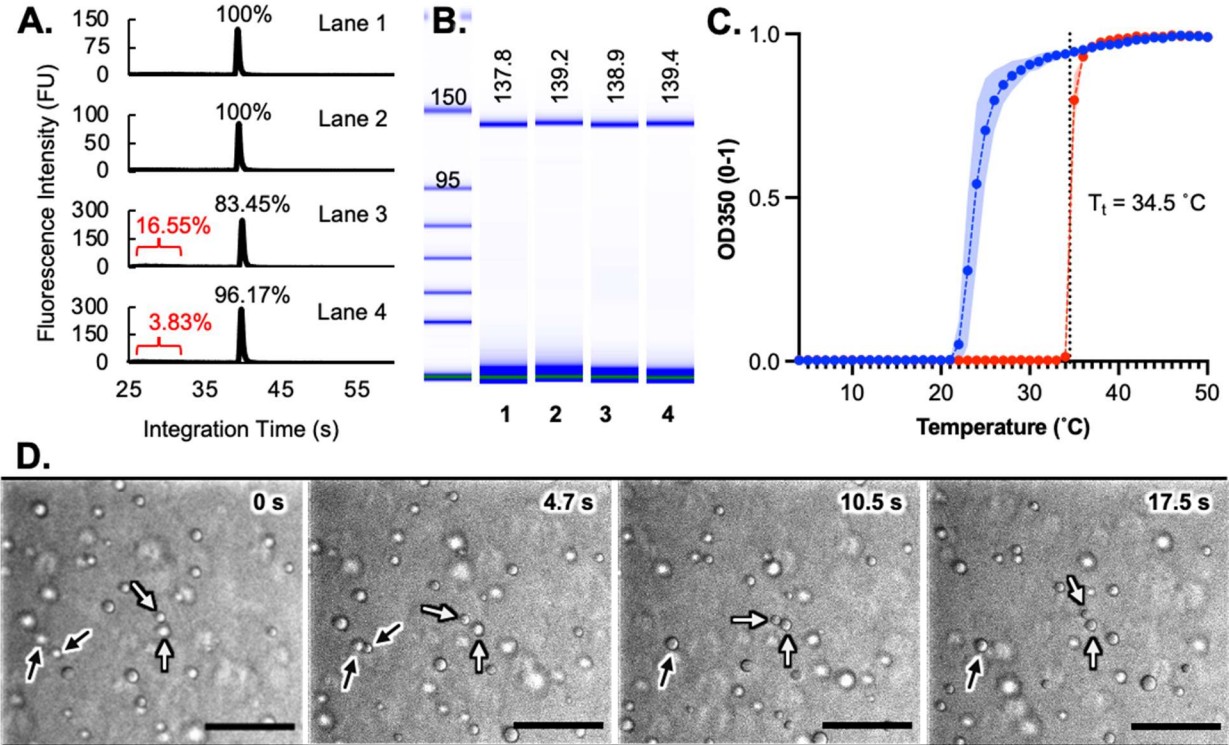

**Figure 4.** ELP-$L_4Y_4$ was expressed, purified, and characterized for turbidimetry and particle formation. After three rounds of ITC, each ELP batch was assessed for purity via an Agilent Bioanalyzer, where the red brackets indicate where impurities were detected. (**A**) Purity was assessed and reported as a percentage by an area under the curve analysis of the fluorescent signal, and the molecular weight was assessed by associating integration time with an Agilent protein ladder. (**B**) These data were used to simulate a PAGE gel where all values are displayed in kDa. (**C**) Purified ELP was also assessed for turbidimetry, where red indicates heating and blue indicates cooling at 1 °C/min. Red and blue shading indicates the standard deviation among three separate batches of ELP. (**D**) Phase-separated ELP-$L_4Y_4$ was visualized using phase contrast at 20× magnification where the scale bar is 20 μm. The time-lapse shows both coalescing (black arrows) and non-coalescing (white arrows) particles.

After dialyzing and freeze-drying, ELP-$L_4Y_4$ batches with >95% purity were resuspended in PBS at 50 μM and evaluated for turbidimetry. The turbidimetry of three batches shows a $T_t$ = ~34.5 °C in Figure 4C. This complements previous literature that showed a $T_t$ of ~37 °C at 25 μM of ELP-$Y_4$ and ~28.5 °C for 25 μM ELP-$Y_6$ [36]. This indicates that the Y-bearing block in ELP-$L_4Y_4$ is acting as the hydrophobic block that allows for the particle assembly of the bulk ELP when the solution temperature is raised to physi-

ological temperatures. To assess microparticle formation and stability, phase-separated ELP-$L_4Y_4$ was observed at 37 °C on a confocal microscope at 20× magnification under phase contrast. While Figure 4D shows this ELP to separate into 1–3 μm diameter particles, these particles are only partially stable as coalescence still occurs (black arrows). However, some microparticles resisted coalescence and appeared to "bounce" off each other (white arrows). A full video of these behaviors is provided in the Supplemental Material (Video S1). Regardless, the observed coalescence is an indicator of the eventual formation of a bulk hydrogel aggregate. In the experimental setup, we noticed that water from the humid setup was drawn into the ELP sample, reducing ELP concentration and elevating the $T_t$ above 37 °C; because of this, ELP assembly was difficult to visualize after ~20 min due to re-solubilization. We plan to address this challenge in future studies and visualize any formation of a hydrophilic shell due to the leucine-bearing block. While we did follow the previous literature's design constraint of the pentapeptide ratio between hydrophilic and hydrophobic blocks (between 1:2 and 2:1) [12], we believe that ELP-$L_4Y_4$ was either too large of an ELP for micelle formation or 50 μM ELP was higher than the critical micelle concentration. Multiple studies that produced ELP micelles determined that the critical micelle concentration of various ELPs ranges between 1–10 μM [11,13,21,37]; however, decreasing the concentration of ELP-$L_4Y_4$ may elevate the $T_t$ above physiological conditions, given that 50 μM ELP already had a $T_t$ close to 37 °C. These studies also used ELPs composed of 100–160 pentapeptide repeats, whereas ELP-$L_4Y_4$ is considerably larger at 240 repeats.

## 3. Discussion

The desired ELP for this study was a di-block ELP for the formation of temperature-sensitive microparticles, where half of the ELP would undergo hydrophobic phase separation at physiological temperatures (<37 °C) and the other half of the ELP would remain soluble, creating a hydrophilic shell that would resist bulk hydrogel formation due to the coalescence of hydrophobic domains. We specifically wanted our hydrophobic block to contain tyrosine (Y) guest residues, which may not elicit the reversible phase separation of ELPs due to their hydrophobicity [38]. Instead, Ingrole et al. demonstrated that the reversible and physiological phase separation of Y-bearing ELP pentamers was possible by combining A (alanine)-bearing pentamers with Y-bearing pentamers in a 4:1 ratio, respectively [36]. Thus, we wanted to re-create this ELP as our hydrophobic block where the primary amino acid sequence is $[(VPG\mathbf{A}G)_2-(VPG\mathbf{Y}G)_1-(VPG\mathbf{A}G)_2]_n$, where $n = 24$ to elicit physiological phase separation [36]. For the hydrophilic ELP block, we decided to maintain the 4:1 ratio of A:X where we selected leucine (L) residues as X, which should not permit physiological phase separation at the same pentapeptide length due to its increased hydrophilicity compared to Y [38]. We decided to maintain a 1:1 ratio for the hydrophilic/hydrophobic block size as previous literature showed diblock ELPs to form stable micelles when block ratios lie between 2:1 and 1:2 [12]. Thus, our final desired ELP sequence is $[(VPG\mathbf{A}G)_2-(VPG\mathbf{Y}G)_1-(VPG\mathbf{A}G)_2]_{24}$—$[(VPG\mathbf{A}G)_2-(VPG\mathbf{L}G)_1-(VPG\mathbf{A}G)_2]_{24}$ for a total of 240 VPGXG repeats with a theoretical mass of 94.97 kDa. The method shown in this study was capable of producing a gene encoding this desired peptide. Uniquely, the method used in this study does not elicit any modifications to the plasmid before gene synthesis and insertion; therefore, we considered the term "plasmid reconstruction" in PRe-RDL to be misleading for the modified technique. Instead, we termed this method RDL after gene termini modification or GTMs-RDL since all required design criteria are encompassed at the ends of the ELP gene rather than the specific plasmid.

Through multiple iterations of GTMs-RDL and transformation of the final gene to pET19b, we ruled out plasmid size (Figure 3B), bacterial growth rate (Figure S3), and total GC content (65% vs. 63%) as significant factors for altered plasmid yields. We speculate that the plasmid to ELP gene ratio may play a role in determining the plasmid yield by altering plasmid maintenance. This notion is supported by the fact that 39% of the pET19b plasmid is composed of the ELP gene (Figure 3D), whereas 55% of the pBluescript

II SK(+) plasmid is composed of the ELP gene (Figure 3C). However, investigating this mechanism requires alternative GC-rich genes and more plasmids of varied sizes to be assessed, which lies outside the scope of this study. This will need to be investigated in future studies. Regardless, the observed decrease in the plasmid yield of pBluescript II SK(+) with increasing ELP gene size demonstrates a limitation on the size of an ELP gene that can be constructed with the GTMs-RDL method.

While Figure 2 proves the success of the GTMs-RDL methodology, the plasmid yields in Figure 3 indicate that the GTMs-RDL did not improve upon the originally published PRe-RDL method that used both an expression plasmid and strain for polymerizing ELP genes. In fact, a single miniprep of the pBluescript II SK(+) plasmid with the final octamer ELP gene yielded enough plasmid in TOP10 (752.5 ± 371.4 ng) for only a single RE digest reaction, whereas the pET19b plasmid miniprep yielded enough plasmid in BL21 (11,031 ± 1198 ng) for multiple digest reactions; we desired a minimum of 500 ng per digest reaction. However, to our knowledge, this is the first time plasmid yields have been reported and compared for a large GC-rich ELP gene (3600 base pairs) in both a high and medium copy number plasmid. This is also the first indication of utilizing the BL21 expression strain for an RDL method. However, ELP-$Y_1$ was shown to have better plasmid maintenance in other bacterial strains, such as pLysS and SoluBL21, though this may not remain true using the significantly larger ELP-$L_4Y_4$ gene [39]. Together, these data indicate that the use of an expression bacterial strain and the minimization of the high GC gene/plasmid ratio may provide the best results for improving ELP-gene plasmid yield in future studies. Outside of the plasmid yield, this study also provided proof of evidence for the use of the FastDigest line of enzymes, which lends the small advantage of a faster workflow over the enzyme library used in the PRe-RDL method.

## 4. Materials and Methods

A full list of enzymes, culture media, purification kits, and chemicals can be seen in Table S1 in the Supplemental Material.

### 4.1. GTMs-RDL of ELP Genes in pBluescript II SK(+)

General molecular biology techniques were used to perform GTMs-RDL, as shown in Figure 1. These basic techniques can be simplified through the use of five broad steps, including (1) RE digestion, (2) digested fragment purification, (3) the ligation of digested fragments, (4) the transformation of ligated plasmids into *E. coli*, (5) and the extraction of expanded plasmid. For step 1, two batches of each ELP monomer, 1–2 μg of DNA each, were digested with either BpiI + AdeI or BmsI + AdeI + PagI for 15 min following the FastDigest manufacturer protocol. PagI was used in the second reaction to minimize overlapping DNA bands in step 2. After digestion, the REs in both reactions were denatured at 80 °C for 15 min. For step 2, both RE reactions were then run through a 0.95 wt% agarose gel containing 1X SYBR™ safe DNA gel stain in TAE buffer for 30 min using a MyGel™ Mini Electrophoresis System (Edison, NJ, USA). A VWR™ blue light transilluminator (Radnor, PA, USA) was then used to cut out the top bands of DNA from both reactions, and DNA was extracted using a DNA extraction kit. For step 3, extracted DNA fragments were then combined and ligated using the T4 Rapid DNA ligation kit for at least 5 min to produce pBluescript II SK(+) plasmids with ELP dimers, ELP-$Y_2$ and ELP-$L_2$. For step 4, these new plasmids were then transformed into One Shot™ TOP10 chemically competent *E. coli*, expanded in S.O.C. media for 1 h, plated onto 100 μg/mL ampicillin LB agar plates, and incubated at 37 °C overnight. Finally, for step 5, 4–6 colonies were then pulled from each plate the next day and expanded in 5 mL of LB medium for 16 h at 37 °C. Furthermore, 20% glycerol stocks were then prepared for each mini-culture, and the new plasmids were isolated using a plasmid mini-prep kit. To confirm the success of GTMs-RDL, 1 μg of each isolated plasmid sample was digested with NcoI and BamHI and run through an agarose gel using the above protocols to confirm the DNA fragment size. The above five steps were repeated to convert the dimer ELP genes to tetramer ELP genes, ELP-$Y_4$ and ELP-$L_4$. This

protocol was repeated with the tetramer ELP genes to yield the di-block ELP gene octamer, ELP-$L_4Y_4$. PagI was not needed in these subsequent RE reactions as the DNA bands of interest did not overlap with any other bands, always remaining the top band in each gel.

When the final ELP gene was produced in pBluescript II SK(+), it was moved to a pET expression plasmid in step 1 via the NcoI + BamHI + PagI RE reaction of the ELP-$L_4Y_4$ pBluescript II SK(+) plasmid, and a NcoI + BamHI reaction of the pET 19b plasmid followed the FastDigest protocol specified above. Once again, PagI was used in the first reaction to prevent any overlapping DNA bands with the octamer gene. For step 2, the top DNA bands from each reaction were then extracted and ligated following the above protocols for step 3, and the new pET19b plasmid containing the ELP-$L_4Y_4$ gene was transformed into BL21(DE3) chemically competent *E. coli* for step 4. After expanding plated colonies in LB media with 100 µg/mL ampicillin, preparing 20% glycerol stocks, and extracting expanded plasmid for step 5, another digestion was performed with NcoI + BamHI to confirm the successful ligation of ELP-$L_4Y_4$ into pET19b. pBluescript II SK(+) and pET19b plasmids were also sent for sequencing using the primers listed in Table 1.

**Table 1.** Primers for sequencing ELP genes in both pBluescript II SK(+) and pET19b plasmids.

|  | **pBluescript II SK(+)** | **pET19b** |
| --- | --- | --- |
| sense: | ggg aac aaa agc tgg agc t | acg act cac tat agg gga att gt |
| antisense: | ggg cga att ggg tac cg | acc cct caa gac ccg ttt ag |

*4.2. ELP Gene Expression and Purification*

To express the final ELP gene, a glycerol stock of the pET19b plasmid containing the ELP-$L_4Y_4$ gene in BL21(DE3) *E. coli* was used to inoculate a 25 mL LB medium culture with 100 µg/mL ampicillin in a 125 mL Erlenmeyer flask. This culture was maintained in a ThermoFisher Scientific Model 420 incubated shaker (Waltham, MA, USA) at 225 rpm for 16 h at 37 °C and then added to a 1 L LB or TB medium culture with 100 µg/mL ampicillin in a baffled 2.8 L Fernbach flask. After an additional 3 h of growth in the Fernbach flask, expression was induced with isopropyl-beta-D-thiogalactoside (IPTG). After 4 h of expression, *E. coli* were condensed into a pellet using a centrifuge at 2000× *g*, resuspended with 20 mL of phosphate-buffered saline (PBS), sonicated with a Biologics Inc. Model 300 V/T ultrasonic homogenizer (Manassas, VA, USA) for 20 min at 40% power and 30% "on" cycles on ice, and stored at −20 °C until ready for ITC purification.

To purify expressed ELP from bacterial lysates, ITC was performed using previous protocols [26]. ITC began with the addition of 2 mL of 10% PEI (polyethylenimine) to each thawed lysate, followed by centrifugation at 16,000× *g* for 10 min at 4 °C in round-bottom 50 mL tubes, and then the disposal of the pellet containing PEI-bound genetic contaminants. The collected supernatant was incubated in a ProBlot hybridization oven (Cary, NC, USA) for at least 30 min at 60 °C to denature bacterial proteins, cooled back to 4 °C, centrifuged again at 16,000× *g* for 10 min at 4 °C in a 50 mL round-bottom tube, and finally the pellet with denatured *E. coli* proteins was discarded. The supernatant from this step was incubated at 40 °C in the Pro-Blot incubator for at least 15 min, and NaCl was added to 2.5 M to ensure phase separation of all ELP. After inverting the tube to mix in all the salt, this supernatant was centrifuged at 16,000× *g* for 10 min at 40 °C in a 50 mL round-bottom tube and the supernatant was discarded. The pellet from this hot spin was then resuspended in 3 mL of PBS and transferred to three 2 mL centrifuge tubes. The resuspended ELP was then further purified by performing multiple back-to-back 4 °C/40 °C centrifugations for 10 min each in order to reversibly phase separate the ELP and remove both soluble and insoluble contaminants by discarding the pellet from cold spins and the supernatant from hot spins. Before each cold spin, trituration was usually required to fully resuspend the hot spin pellet containing ELP. Each batch of ELP went through three of these hot–cold cycles for purification [26].

After purification, each batch of ELP was dialyzed in DI (deionized) $H_2O$ at 4 °C using 500–1000 Da MWCO dialysis cassettes. Dialyzed ELP was then added to a tared 15 mL conical tube, frozen at $-80$ °C, and then lyophilized on a Labconco FreeZone 2.5 L -84C benchtop freeze dryer system (Kansas City, MO, USA) for at least 24 h. After lyophilizing, the mass of ELP was measured to assess the expression yields.

*4.3. ELP Characterization*

The ELP purity and size were assessed on an Agilent 2100 bioanalyzer (Santa Clara, CA, USA) using HS protein electrophoretic chips and following the manufacturer protocol for fluorescent labeling and chip loading. Briefly, 4.5 µL of each dialyzed ELP batch was added to 0.5 µL of the supplied labeling buffer, labeled for 30 min, diluted 1:200 in DI $H_2O$, denatured for 5 min at 95 °C in non-reducing conditions (water added to manufacturer denaturing solution), and then loaded onto the HS protein chip. After running the chip, the bioanalyzer provided a report for each sample that included the percentage of the total area under the curve (AUC) for each protein detected during the sample run. This AUC% was used to quantify the purity of each ELP batch. Each detected protein was also compared to the manufacturer's ladder to provide an approximate weight in the report. The full report for the ×4 batches used in this study can be seen in the Supplemental Material (Figure S3).

Three batches of ELP-$L_4Y_4$ were used to evaluate the turbidimetry of the final ELP product using a U-2910 double-beam Hitachi spectrophotometer (Tokyo, Japan) with a Quantum Northwest TC 1 temperature controller (Liberty Lake, WA, USA). PBS was added to each ELP batch to yield a 50 µM solution that was added to a 10 mm path-length polystyrene cuvette. The sample cuvette was added to the sample holder, and a cuvette with PBS was added to the reference holder. Both cuvettes were allowed to cool to 4 °C using the temperature controller under a continuous flow of $N_2$ at 1–2 LPM. After reaching 4 °C, cuvettes were ramped to 50 °C at 1 °C/min increments, where a 350 nm measurement was taken at each ramping step. Subsequently, each cuvette was cooled back down to 4 °C at 1 °C/min with 350 nm measurements at each temperature step. The inflection point of the heating slope indicates the lower critical solution temperature or transition temperature ($T_t$), and the cooling slopes verify the reversibility of ELP-$L_4Y_4$ phase separation.

To evaluate the particle size of phase-separated ELP-$L_4Y_4$ at 50 µM, a video at 20× magnification was taken on a Keyence BZ-X710 All-in-One Fluorescence Microscope (Osaka, Japan) under phase-contrast illumination in a temperature-controlled chamber. Furthermore, 20 µL of ELP was added to a 2-well multiwell chambered coverslip, and the coverslip was heated to 37 °C using a TOKAI HIT heating stage top incubator (Fujinomiya, Japan). It should be noted that the sample volume on the coverglass was not isolated from the stage-top incubator, and moisture appeared to be drawn out of the humid, temperature-controlled environment and into the sample after ~20 min. Therefore, a short video was captured to observe the behavior of microparticles immediately after physical phase separation occurred. The full video can be downloaded from the Supplemental Material (Video S1).

**5. Future Direction and Conclusions**

While this study demonstrates the success of GTMs-RDL, we wanted to note that both N- and C-terminus modifications are also possible following the RE digest reactions shown in Figure 5. This is important to note since many applications of ELP particles in the biomaterials and drug delivery field rely on the implementation of specific amino acid sequences, such as adhesive moieties, cell-penetrating peptides, and therapeutic proteins [11–14,25,34,35]. We expect to integrate the procedure outlined in Figure 5 in future works to adapt our ELP-$L_4Y_4$ for biomedical applications.

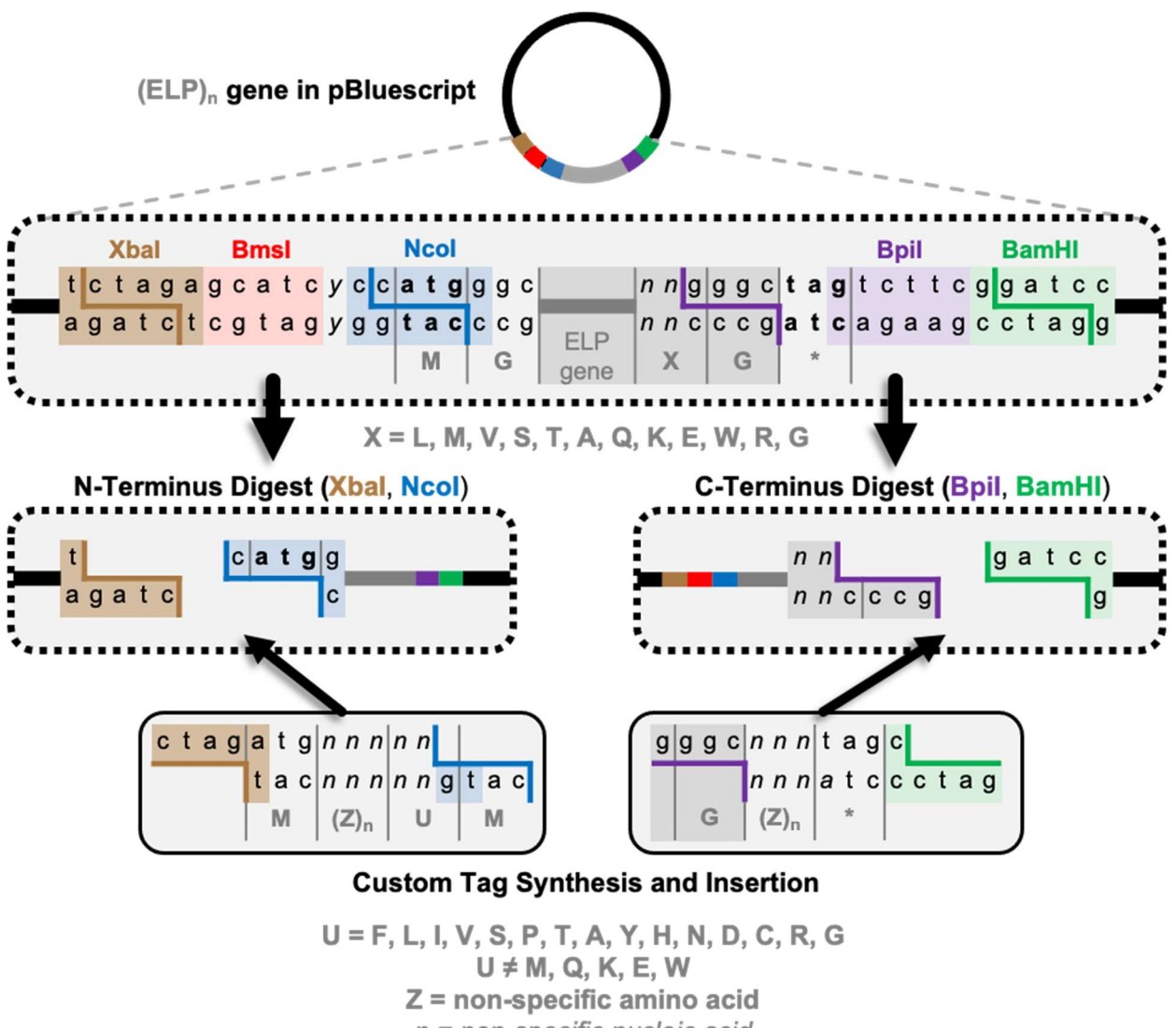

**Figure 5.** After completing the last round of GTMs-RDL, modification of the gene termini can be performed in a single digest reaction to integrate a short therapeutic or bioactive moiety. Due to the design criteria for GTMs-RDL, modification at the N-terminus will always result in methionine and a specified amino acid (U) between the inserted fragment and the ELP gene. Modification of the C-terminus does not have any limitations on the encoded amino acid sequence. The ELP gene is shown in gray. The enzyme binding sites are highlighted and enzyme cut sites are outlined in brown for XbaI, red for BmsI, blue for NcoI, purple for BpiI, and green for BamHI. * indicates the stop codon sequence (tag).

In this study, we employed a modified version of PRe-RDL that we termed RDL by gene termini modifications or GTMs-RDL. This method was specifically designed using the high copy number plasmid pBluescript II SK(+) to improve plasmid yields for multiple rounds of GTMs-RDL. Because this method is not performed in a pET expression plasmid, it is considered the most appropriate for producing a final ELP gene, requiring multiple rounds of GTMs-RDL to produce a single ELP instead of a library of ELPs of various sizes. Using this modified method, we successfully produced a 240 pentapeptide ELP diblock gene, encoding tyrosine guest residues in the hydrophobic block and leucine guest residues in the hydrophilic block. While this method successfully produced a large, GC-rich ELP gene, we discovered that the high copy number maintenance plasmid resulted in lower plasmid yields when compared to the expression medium copy number plasmid, regardless of bacterial strain. Additionally, the plasmid yields from the BL21 expression strain were

higher than the TOP 10 maintenance strain regardless of plasmid type. While these findings are counter-intuitive, these data provide insight that the GC-rich gene/plasmid ratio may be a more important factor than either the total plasmid size or total plasmid GC content for providing larger plasmid yields of large, GC-rich ELP genes. Future work can further investigate this gene/plasmid ratio by adapting the GTMs-RDL method to other plasmids. The final ELP gene was expressed in BL21 *E. coli*, and ELPs were successfully purified via ITC and shown to undergo semi-stable microparticle formation at a concentration of 50 μM. Our future work will utilize the GTMs-RDL method for other ELP genes, add various tags to the gene termini, and explore ELP-$L_4Y_4$ phase-separation behavior further for micelle assembly as well as biomedical applications.

**Supplementary Materials:** The following supporting information can be downloaded at: https://www.mdpi.com/article/10.3390/synbio2020010/s1, Table S1: List of materials; Figure S1: Representative images of agarose gels without annotations; Figure S2: Transformed colonies on plates; Figure S3: Optical density measurements of bacterial strains with different plasmids after 16 h. Figure S4: Autogenerated reports from bioanalyzer; Video S1: Video of particle motion.

**Author Contributions:** Conceptualization, D.W.N. and A.C.; methodology, D.W.N. and Y.S.; formal analysis, D.W.N.; investigation, D.W.N.; writing—original draft preparation, D.W.N.; writing—review and editing, D.W.N., A.C. and R.J.G.; visualization, D.W.N.; supervision, R.J.G.; funding acquisition, R.J.G. All authors have read and agreed to the published version of the manuscript.

**Funding:** Funding for this work was provided by Veteran Affairs I01 grant (I01RX003502), Veteran Affairs I21 grant (I21RX004406), and the New York State Spinal Cord Injury Research Board (NYSCIRB) Institutional Support Grant (#C38335GG) awarded to R.J.G., NIH T32 Grant (#AG057464) and the Center for Disability Services' Health Innovations Incubator and Technology (HII-Tech) Center Student Fellowship awards supporting D.W.N., and NSF funding (NSF ITE-2236099) supporting A.C.

**Institutional Review Board Statement:** Not applicable.

**Data Availability Statement:** All data will be made available upon request.

**Acknowledgments:** The authors would like to acknowledge the Genomics Core Facility and the Microbiology and Fermentation Core Facility run by Yang Bai and Joel Morgan, respectively, at the Rensselaer Polytechnic Institute for material acquisition and equipment usage.

**Conflicts of Interest:** The authors declare no conflicts of interest.

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
