# Peer review of "Construction of an Elastin-like Polypeptide Gene in a High Copy Number Plasmid Using a Modified Method of Recursive Directional Ligation"

_2674-0583, doi:10.3390/synbio2020010_

Round 1
Reviewer 1 Report
Comments and Suggestions for Authors
Dear authors,
In the paper " Construction of an elastin-like polypeptide gene in a high copy 2 number plasmid using a modified method of reverse directional ligation," D. W. Nelson et al. discuss the construction of an elastin-like polypeptide (ELP) gene using a modified method of reverse directional ligation (PRe-RDL) to achieve precise control over the number of repeat units in the ELP. The study highlights successfully producing a diblock ELP gene with 240 pentapeptide repeats using a high copy number plasmid and Type IIS restriction enzymes. The paper underscores the potential of ELPs as biomaterials with tunable properties for various biomedical applications and the significance of innovative cloning techniques. This paper presents sufficient content for publication in SynBio. I recommend publication if the following point is noted and improved.
L462 Figure 4D
In Figure 4D, other particles surrounding the target particle also change simultaneously. It is easy to confirm whether they are fused in the video. However, noticing the movie's existence is difficult because only the annotation "Video S1" is in Materials and Methods.
Author Response
Thank you for your comment. We added a reference to Video S1 to the Results and Discussion section 3.2.
Reviewer 2 Report
Comments and Suggestions for Authors
In the manuscript „ Construction of an elastin-like polypeptide gene in a high copy number plasmid using a modified method of reverse directional ligation“ the Authors present their findings concerning optimizing large ELPs synthesis. The chosen construction approach is worth, however several issues should be addressed:
Major remarks:
1. When describing the aim of the work performed, the Authors start with negative part – seemingly explaining their choices as if those choices could be considered incorrect. I would advise to rephrase the formulation of the aim stressing the parts that are innovative first, and only then, if need be, addressing the possible shortcomings or limitations.
2. The introduction part would be greatly advanced by introducing the existing schemes of Pre-RDL based ELP multimer construction. This could make understanding the unique part of this project easier.
3. Method information, especially very detailed one, is repeated twice, in the Methods and in the Results. It should be avoided. Also, the explanation for the methods chosen should be moved to Results. Altogether, I would suggest the Authors 1) clarify the information that is given in the Methods section. Bacterial growth conditions should be given in separate chapter and not repeated several times, as well as other methods (e.g. RE conditions etc.) can be separated into shorter and clearer chapters. 2) Results should not repeat minor details already given in the Methods, but instead explain the Authors’ choices and present their background. The criteria of design would fit better in the Result section, as it was the Authors’ work.
4. The constructs contain long repetitive sequences, propagated in E. coli. Even the bioanalyzer data gives different protein sizes to the predicted ones. Are you sure that the constructs are of the right size, and no unexpected homologous recombination, or other events, have happened, and the final constructs are correct?
5. The Authors claim in lines 346-7 that electroporation could be used as a better method. No reference to such claim is made, and no indication that the Authors have tested that.
6. The Authors have noticed the decrease of plasmid yield. However, it is unclear if the growth of bacteria was also decreased.
7. The Authors claim, that BL21 is a better strain to use, however, B strains tend to grow faster than K12 strains. Perhaps the issue is (as mentioned in the Q6) that the growth rate of bacteria is changed due to large plasmid load on metabolism?
8. Lines 396-7 there is not enough information presented to base the Authors’ claim of GC content playing a major role in plasmid yield. Also, as mentioned before, it is unclear if the plasmid copy number is to blame, o the slower growth of bacteria.
Minor remarks:
1. Throughout the manuscript the E. coli name is written incorrectly – please address.
2. Why does the word order in the abbreviation “reverse direction ligation by plasmid reconstruction (PRe-RDL)” seem to be in the opposite order?
3. Line 18 “(VPGXG)n” n should be in subscript.
4. Some changes in academic language advised – line 92 should not say that REs “care”, line 249 should not say “cooked”.
5. Abbreviations PEI, DI are not explained in the text.
6. Lines 312-3 it is unclear in which order the constructs were made, which parts were synthesized, and which ones inserted.
Author Response
Major remarks:
- When describing the aim of the work performed, the Authors start with negative part – seemingly explaining their choices as if those choices could be considered incorrect. I would advise to rephrase the formulation of the aim stressing the parts that are innovative first, and only then, if need be, addressing the possible shortcomings or limitations.
Thank you for helping identify a misleading premise for describing the project goal in the introduction. The text was edited to highlight this project’s innovation first as follows:
“In this article, we aimed to produce an ELP gene that was composed of a previously published ELP polymer, with already-defined properties but modified with an appended hydrophilic ELP block. Thus, we developed a novel plasmid engineering scheme to use methods similar to PRe-RDL in a high copy number plasmid, pBluescript II SK(+), which lends the possible benefit of providing higher plasmid yields from bacterial cultures.”
- The introduction part would be greatly advanced by introducing the existing schemes of Pre-RDL based ELP multimer construction. This could make understanding the unique part of this project easier.
We apologize for any confusion, but the Pre-RDL method is described in detail in the second paragraph of the introduction (Lines 88-112)
- Method information, especially very detailed one, is repeated twice, in the Methods and in the Results. It should be avoided. Also, the explanation for the methods chosen should be moved to Results. Altogether, I would suggest the Authors 1) clarify the information that is given in the Methods section. Bacterial growth conditions should be given in separate chapter and not repeated several times, as well as other methods (e.g. RE conditions etc.) can be separated into shorter and clearer chapters. 2) Results should not repeat minor details already given in the Methods, but instead explain the Authors’ choices and present their background. The criteria of design would fit better in the Result section, as it was the Authors’ work.
The first paragraph of section 2.1 was the rationale included in the methods chapter and has now been moved to the first paragraph of the results and discussion chapter in section 3.1. We attempt to clarify methods in section 2.2 by breaking down the methodology into 5 generalized steps. We did not want to separate these protocols into different chapters as they are interdependent and, together, compose the proposed GTMs-RDL methodology. If a reader wants to repeat this method, we improved the text to identify each of the five steps in both the methods and results chapters. We included the following texts:
“These basic techniques can be simplified by five broad steps including 1) RE digestion, 2) digested fragment purification, 3) ligation of digested fragments, 4) transformation of ligated plasmids into E. coli, 5) and extraction of expanded plasmid.”
“For step 1…”
“…following steps 4 from section 2.2”
“Step 5 was then performed with…”
We hope that these edits help clarify the method information and remove some of the redundancy in this manuscript.
- The constructs contain long repetitive sequences, propagated in E. coli. Even the bioanalyzer data gives different protein sizes to the predicted ones. Are you sure that the constructs are of the right size, and no unexpected homologous recombination, or other events, have happened, and the final constructs are correct?
Because all plasmids were sequenced using a 3rd party vendor, we know that the final ELP gene being expressed is correct. Regarding the expressed ELP, we know from previous literature that it is expected for ELPs to appear much larger than their actual size in an electrophoretic analysis (reference 26). Additionally, the expressed ELP displays a Tt that very closely resembles previous literature indicating that the hydrophobic block is behaving similarly to its theoretical size (reference 35). Taken together we successfully created an ELP with a predictable phase separation that undergoes a semi-stable microgel formation, which was one of the smaller goals in his study as specified at the beginning of section 3.1. Lastly, this is a large peptide that lies outside of the range of commercially available standards for mass spectrometry (specifically MALDI-TOF). As such we are currently investigating this size determination in future work.
- The Authors claim in lines 346-7 that electroporation could be used as a better method. No reference to such claim is made, and no indication that the Authors have tested that.
While there is prior literature that electroporation can improve transformation efficiency, we decided to remove this text as electroporation would likely fail in this application due to the presence of salts from the ligation reaction. Because this paper is focused on a method that always requires ligation (x1 round of RDL or transferring the gene to a pET plasmid), we thought that this comment was not beneficial. Thank you for helping bring this to our attention.
- The Authors have noticed the decrease of plasmid yield. However, it is unclear if the growth of bacteria was also decreased.
This is an excellent point that we decided to address by including an additional supplemental figure showing the optical density of bacterial cultures used for Figure 3B. This figure shows that there was no significant difference in bacterial population after the 16-hour growth period. The following text was included in the results section:
“Additionally, there was no difference in cell density after 16 hours, as indicated by OD600 measurements in Figure S3, which implies that the plasmids did not significantly affect bacterial growth.”
- The Authors claim, that BL21 is a better strain to use, however, B strains tend to grow faster than K12 strains. Perhaps the issue is (as mentioned in the Q6) that the growth rate of bacteria is changed due to large plasmid load on metabolism?
Please see the above comment. Because there was no difference in final cell density, we speculate that plasmid maintenance may be affected by the selected plasmid containing the full ELP gene. Because this study is focused on the establishment of a new plasmid engineering scheme, we feel that investigating plasmid maintenance is outside the scope of this project. However, we included the following text to address this comment:
“Taken together, we have ruled out plasmid size (Figure 3B), bacterial growth rate (Figure S3), and total GC content (65% vs 63%) as significant factors for altered plasmid yields. We speculate that the plasmid to ELP gene ratio may play a role in determining plasmid yield by altering plasmid maintenance. This notion is supported by the fact that 39% of the pET19b plasmid is composed of the ELP gene (Figure 3D) whereas 55% of the pBluescript II SK(+) plasmid is composed of the ELP gene (Figure 3C). However, investigating this mechanism requires alternative GC-rich genes and more plasmids of varied sizes to be assessed, which lies outside the scope of this study. This will need to be investigated further in future studies. Regardless, the observed decrease in plasmid yield of pBluescript II SK(+) with increasing ELP gene size demonstrates a limitation on the size of an ELP gene that can be constructed with the GTMs-RDL method.”
- Lines 396-7 there is not enough information presented to base the Authors’ claim of GC content playing a major role in plasmid yield. Also, as mentioned before, it is unclear if the plasmid copy number is to blame, o the slower growth of bacteria.
It was not our intention to claim that GC content is the dominating factor for altered plasmid yields, but rather we wanted to speculate that the ratio of the gene (which is super GC-rich) to plasmid size may be a significant factor for these differences, especially since we ruled out overall GC content, plasmid size, and final cell density as parameters that affect plasmid yields. This size ratio may be a factor for altered plasmid maintenance. To help explain this rationale, we decided to include a GC map for the pET plasmid in Figure 3D. We also changed some of the text to emphasize that we are hypothesizing or speculating about this potential mechanism. As stated above, pursuing this hypothesis lies outside the scope of this study but we still wanted to acknowledge the limitation of the GTMs-RDL method is increasing gene size.
Minor remarks:
- Throughout the manuscript the E. coli name is written incorrectly – please address.
If this comment is referring to the capitalization of c in coli, then this has been addressed throughout the document.
- Why does the word order in the abbreviation “reverse direction ligation by plasmid reconstruction (PRe-RDL)” seem to be in the opposite order?
We also found this abbreviation to be odd, but McDaniel et al. designated their methodology with this acronym, so we did not want to change it in this paper.
- Line 18 “(VPGXG)n” n should be in subscript.
Thank you for catching this error. This has been corrected in the abstract.
- Some changes in academic language advised – line 92 should not say that REs “care”, line 249 should not say “cooked”.
Thank you for catching this terminology. The sentences have been replaced as follows:
“This allows for the cleavage of any DNA sequence as Type IIS REs are only specific about the cut distance from their recognition site and not the cut sequence.”
“The collected supernatant was incubated in a ProBlot hybridization oven for at least 30 minutes at 60ËšC to denature bacterial proteins…”
- Abbreviations PEI, DI are not explained in the text.
These acronyms are now defined in the text (lines 231 and 249).
- Lines 312-3 it is unclear in which order the constructs were made, which parts were synthesized, and which ones inserted.
To address reviewer comments from above, this text was modified:
“Using the five design criteria outlined in section 2.1, ELP-Y1 and ELP-L1 genes were synthesized and inserted into pBluescript II SK(+) plasmids using Genscript’s molecular biology services. Following the 5 general steps outlined in section 2.2…”
Round 2
Reviewer 2 Report
Comments and Suggestions for Authors
I want to thank the Authors for addressing my concerns. I would just like to mention, that E. coli should be written in coursive; and the presented Supplementary files are updated at the end of the Manuscript, but not in the separate file.
Author Response
I want to thank the Authors for addressing my concerns. I would just like to mention, that E. coli should be written in coursive; and the presented Supplementary files are updated at the end of the Manuscript, but not in the separate file. We thank the reviewer for their comments. We have italicized E. coli throughout the manuscript. and we have uploaded an updated supplementary document file.